# Stearic Acid/Layered Double Hydroxides Composite Thin Films Deposited by Combined Laser Techniques

**DOI:** 10.3390/molecules25184097

**Published:** 2020-09-08

**Authors:** Ruxandra Birjega, Andreea Matei, Valentina Marascu, Angela Vlad, Maria Daniela Ionita, Maria Dinescu, Rodica Zăvoianu, Mihai Cosmin Corobea

**Affiliations:** 1National Institute for Lasers, Plasma and Radiation Physics, 409 Atomistilor Street, P.O. Box MG-16, 077125 Magurele-Bucharest, Romania; ruxandra.birjega@inflpr.ro (R.B.); valentina.marscu@inflpr.ro (V.M.); angela.vlad@inflpr.ro (A.V.); daniela.ionita@infim.ro (M.D.I.); dinescum@nipne.ro (M.D.); 2Faculty of Chemistry, Department of Organic Chemistry, Biochemistry and Catalysis, University of Bucharest, 4-12 Regina Elisabeta Av., S3, 030018 Bucharest, Romania; rodica.zavoianu@chimie.unibuc.ro; 3National Institute for Research and Development in Chemistry and Petrochemistry, 202 Spl.Independentei, 060021 Bucharest, Romania; mcorobea@yahoo.com

**Keywords:** thin films, laser deposition, layered double hydroxides, stearic acid

## Abstract

We report on the investigation of stearic acid-layered double hydroxide (LDH) composite films, with controlled wettability capabilities, deposited by a combined pulsed laser deposition (PLD)-matrix-assisted pulsed laser evaporation (MAPLE) system. Two pulsed lasers working in IR or UV were used for experiments, allowing the use of proper deposition parameters (wavelength, laser fluence, repetition rate) for each organic and inorganic component material. We have studied the time stability and wettability properties of the films and we have seen that the morphology of the surface has a low effect on the wettability of the surfaces. The obtained composite films consist in stearic acid aggregates in LDH structure, exhibiting a shift to hydrophobicity after 36 months of storage.

## 1. Introduction

The miniaturization tendency related to the last decade of industrial development has led to an increased need for new materials, based on hybrid or composite complex structures, with combined properties that can be tailored. More specifically, organic/inorganic compounds gathering capabilities derived from both their organic and inorganic chemical groups are playing an important role from the practical and theoretical point of view.

Layered double hydroxides (LDHs), also known as hydrotalcites or anionic clays, are a class of host-guest materials that can fulfill the demands listed above. They are defined by the formula
[M^2+^_1*−x*_M^3+^*_x_*(OH)_2_]*^x+^*(A*^n−^*)*_x/n_*·*m*H_2_O(1)
where M^2+^ and M^3+^ are divalent and trivalent metal ions, respectively, and A^n^*^−^* is an exchangeable anion compensating the positive charge of the hydroxide layers [1]. The LDH property that makes them a very appealing material is the adsorption behavior, due to their high layer charge density, their anionic exchange capacity, and their swelling abilities [2].

A broad area of practical applications has been reported until now for layered double hydroxides such as anion exchangers, polymer stabilizers, anion scavengers, catalysts and catalyst supports, adsorbents, photoactive materials, and pharmaceutics [3].

Functional materials can be designed and produced by introducing anions with specific functions in LDH structure. Drugs, vitamins, anti-inflammatory substances have been entrapped between the brucite layers for pharmaceutical uses, pesticides for agriculture [2], polymers for protective coatings and electrochemical devices [4,5,6,7], halocomplexes for catalysis [8], etc. Behaving as an absorbent, LDH can be used for the removal of the contaminant biological materials as Escherichia coli and heavy metals from water [9,10].

Chemical sensors, protective coatings, components in optical and magnetic devices, drug delivery industry, etc. are possible applications for LDHs as thin films. For all these applications, a crystalline controlled structure is needed [11].

Thin films of tens to hundreds of nanometers and organic/inorganic multistructures are mandatorily needed for microelectronics, sensors, and coatings technology. We have previously demonstrated that laser techniques can be employed for the deposition of LDH films or their derived mixed oxides [12,13,14,15], matrix-assisted pulsed laser evaporation (MAPLE), and pulsed laser deposition (PLD) offering the advantages of supplying adherent films with controlled thickness. The conventional wet techniques (e.g., spin coating) cannot offer the thickness and uniformity control needed for applications. Pulsed laser deposition (PLD) has been successfully used in fabrication of both inorganic and organic thin films [16], but in order to have a soft process for organic materials, MAPLE is used as a method capable of producing thin films of materials with unaltered chemical structure [17,18]. Moreover, laser methods can be used for multilayers production and combining MAPLE with PLD, complex heterostructures organic/inorganic can be designed and fabricated [19].

An important aspect to be considered in the production of nanoscale devices, designed for chemical, biological sensing or as protective coating is the surface wettability. It is well-known that two dominant factors play an important role in improving hydrophobicity: surface hierarchical micro/nanostructure and surface low free energy.

In this work, we report on the employing standard pulsed laser deposition and matrix-assisted pulsed laser evaporation for the growth of the composite organic-inorganic layers (stearic acid-LDH). A MAPLE deposition system has been adapted in such a way to provide the possibility of irradiating two concentric targets, with two different laser beams. The PLD target was a pressed LDH pellet placed in the center and the exterior MAPLE target was a frozen solution of stearic acid. The films produced by this combined MAPLE-PLD system are stearic acid/layered double hydroxides (LDH) composite layers, with tunable wettability capabilities.

Long-chain organic acids, such as carboxylic acid, are beneficial for practical application, in particular for corrosion protection due to their strong bond to oxidized surfaces of metals and alloys [20,21,22,23,24,25]. Most of the studies are focusing on the thin film structure and binding of the head group to the surface and few on the durability of the protective layer against atmospheric and aqueous corrosion prolonged exposure.

We have investigated stearic acid as a test molecule because it is a well-known long chain (18 carbon atoms) saturated fatty acid widely found in the nature. We have also examined the long-time storage stability of the composite films hydrophobicity and the role played by the stearic acid waxing agent. The depositions were performed on flat/untreated surface of double side polished Si wafers. In this way, there is no contribution of the morphological architecture of the substrate in the likely increase of the as-deposited films’ hydrophobicity.

The experimental system, beside growing composite thin films, can be used for growing organic/inorganic multi-structures, without opening the reaction chamber, and so avoiding the contamination between different deposition processes.

## 2. Materials and Methods

### 2.1. Materials

The investigated materials are doubled layered hydroxides based on Mg and Al with the atomic ratio Mg/Al = 2.5. The primary powders were prepared by co-precipitation at 40 °C under low supersaturation conditions at pH of 9.5–10 using aqueous solutions of Mg and Al nitrates. The obtained gel was dried at 85 °C for 24 h. All the reagents were commercially available, chemical pure grade chemicals (Merck (Kenilworth, NJ, USA), Fluka (CERN, Meyrin, Switzerland) and Aldrich-Sigma (St. Louis, MO, USA), respectively). The as-obtained powders were pressed as round pellets to be used as targets for the PLD deposition experiments.

### 2.2. Thin Films Deposition

The combined MAPLE-PLD system (Figure 1) consists of a standard MAPLE deposition chamber equipped with a liquid nitrogen cooling system which has been adapted for having two lasers irradiating a complex target. The pressed LDH target is placed in the center of the frozen MAPLE target. A lens with focal distance of 300 mm and a diameter of 100 mm is placed outside the deposition chamber and it is used for focusing both laser beams, one on the surface of PLD target, and one on the surface of the MAPLE target. The PLD target was a pressed pellet of Mg-, Al-based layered double hydroxide with Mg/Al molar ratio of 2.5 (Mg2.5Al), while the MAPLE target consisted in 5 wt% of stearic acid dissolved in water: ethanol (1:1 mass ratio). Two Nd:YAG pulsed lasers working in IR or UV were used to irradiate the complex target. Based on our previous experiments [10,11,12,13,14,15,16], we established the best set of deposition conditions in the range of 0.5–0.6 J/cm^2^ for MAPLE (266 nm wavelength, 10 Hz) and 1.5–3 J/cm^2^ for PLD (266 or 1064 nm wavelength, 3 Hz), depending on the used wavelength. All depositions took place in vacuum (1 × 10^−4^ mbar—starting pressure), with an increase of the pressure during irradiation, due to the solvent evaporation. The number of pulses for irradiating the MAPLE target was 36,000 and for the PLD target was 12,000. The average deposition rate is 0.05–0.1 Å/pulse. Thin films of pure stearic acid were deposited by MAPLE, irradiating a frozen target of 5 wt% of stearic acid. The silicon substrates were kept at room temperature during thin film growth. The target—substrate distance was chosen to be 3.5 cm, with the target rotated during experiments. Thin films of Mg2.5Al were deposited by standard PLD, at 266 nm and 1064 nm incidence wavelength, as a result of 20.000 pulses at 2 J/cm^2^. The thin films were denoted as follows in Table 1.

### 2.3. Characterization

Thin films morphology and roughness were analyzed by atomic force microscopy-AFM (XE-100 type from Park Systems, Suwon, Korea) and scanning electron microscopy-SEM (FEI, model Inspect S50, Eindhoven, The Netherlands) Fourier transform infrared spectroscopy-FTIR (JASCO FTIR 6300 spectrometer, Vienna, Austria) working on transmission mode were used to survey especially the organic component. The structural investigations of the thin films were performed by X-Ray Diffraction at grazing incidence (GI-XRD ω = 0.25^°^) on a Panalytical X’Pert MRD system (Almelo, The Netherlands) (λ_CuKα_ = 1.5418 Å). The crystalline structure of the primary materials was checked by X-ray Diffraction (XRD) using a Bragg-Brentano geometry on a Panalytical X’Pert MPD system (λ_CuKα_ = 1.5418 Å). Contact angle (CA) measurements were performed using a Contact Angle Tensiometer CAM 200 from KSV Instruments, Filderstadt, Germany).

## 3. Results and Discussion

The XRD patterns of the primary materials, commercial stearic acid (St) and as-prepared Mg2.5Al-LDH according to the protocol described previously, are presented in Figure 2. The XRD patterns match the standard references from ICDD XRD database: card no 00-038-1923 for the stearic acid and, no. 01-089-0460 for the layered double structure, hydrotalcite-type (Mg_0.667_Al_0.333_)(OH)_2_(CO_3_)_0.167_(H_2_O)_0.5_, respectively.

XRD patterns (Figure 3) for the films deposited via combined PLD-MAPLE method reveal the deposition of a (00*l*) oriented LDH structure similar to our previous results [10,11,12,13,14,15] and to the present single PLD depositions. The lattice *c* parameter and the coherence length along the c-axis, D_003_, are included in Table 2. The data inspection shows slightly smaller values of the *c* parameter for the combined MAPLE-PLD deposition for both wavelengths in comparison with single PLD depositions due probably to the preference for OH^-^ incorporation instead of larger CO_3_^2−^ interlayer anions. We observed also larger D_003_ values at 1064 nm wavelength PLD deposition in both cases: combined PLD-MAPLE and PLD, respectively. Generally, due to the higher wavelength penetration depth in the target, the 1064 nm wavelength favored the deposition of thicker films [13]. The MAPLE deposition of stearic acid conducted to the formation of an amorphous film.

Generally, the hydrophobicity of a surface requires a combination of roughing surfaces and low-surface-energy materials [21]. Long chain stearic acid provides low-surface-energy coating for different surfaces [20,21,22,23,24,25,26,27].

The morphological aspect of as-deposited films is disclosed in Figure 4. The roughness of surface of the films, as quantified by their root-mean-square (RMS) deviation values, is included in Table 2. The deposition of an organic material as stearic acid gives a smoother surface with a small roughness of 8 nm. We already observed and reported that the deposition of LDHs via PLD at 266 nm wavelength gives granular and dense aspect surfaces with high roughness [12,14], while in the LDH films obtained at 1064 nm wavelength, big aggregates are formed, but by limiting their number their RMS values are considerably smaller [10,14]. The same outcome is obtained for the present films: grainy, compact, and dense aspect, and high roughness for the film deposited by PLD at 266 nm reference Mg2.5AlPLD(266 nm) as well as for the composite film StMAPLE(266 nm)/Mg2.5AlPLD(266 nm) while at 1064 nm for both Mg2.5AlPLD(1064 nm) and StMAPLE(266 nm)/Mg2.5AlPLD(1064 nm), the surfaces are less compact, exposing splashes conducting to smaller roughness.

The contact angle values and the photographs of the water droplets on the surface of the as-deposited films are presented in Figure 5. We measured the contact angle on as-deposited films and after 36-months storage. The contact angles (CA) are around 85° for the flat, but low-energy surface of SA film and for the rough surfaces of the films deposited via PLD at 266 nm both single Mg2.5AlPLD(266 nm) and combined by MAPLE addition StMAPLE(266 nm)/Mg2.5Al-PLD(266 nm). The films exhibiting lower roughness obtained via PLD at 1064 nm in both cases single PLD and in combination with MAPLE are more hydrophilic. After 36 months of storage, all the films except the composite films show a decrease of the CA, probably due to moisture and contaminant exposition. The composite films exhibit a significant increase of CA, the films becoming hydrophobic. Coatings preserving their hydrophobic properties after long-time storage are of great interest for their protective function especially for metallic surfaces. There are several papers reported on the transformation of micro-nanostructured metal and metal/oxides from initially hydrophilic to hydrophobic and even super-hydrophobic by air exposure without any other chemical modifications [28,29,30,31,32,33,34]. The explanations are quite diverse. Xiao et.al claimed that after one-month storage, a dense oxide film is formed on Co substrates lowering their surface energy [28]. The transition from super-hydrophilic to super-hydrophobic of a Ni micro-nano cones array surface were attributed by Geng et al. to the formation of NiO during the two weeks storage [29]. For Wang et al. [30], the increase of the contact angle to super-hydrophobicity of nanostructured CuO film during three weeks of air exposure might be caused by the physical adsorption of oxygen molecules on the surface. S. Korsand et al. [31,32,33] attributed the wettability transition from a super hydrophilic nature to a super-hydrophobic one of Ni films and Ni-Co films mainly to their very particular hierarchically micro-nano structure, which, by allowing air to be trapped in rough surfaces during the two weeks exposure to air, will form convex surface between the interface of liquid and air. For the Ni-Co alloy coatings, the surface adsorption of air-borne hydrocarbons [32,33] is also considered similar to Li et al. for several metal, Cu, Ni, Au, etc. as a cause of increased super-hydrophobicity by their effect of reducing the surface free energy [34]. After reaching super-hydrophobicity the Ni-Co alloy coatings exhibit long time durability, the coatings being air exposed more than 4 months [32]. Mg2.5AlPLD(266 nm) film exhibiting higher roughness (RMS) and a morphological-like appearance as the composite StMAPLE(266 nm)/Mg2.5AlPLD(266 nm) (Table 2 and Figure 4) undergoes no increase of CA in 36 months of storage, while the composite film becomes hydrophobic. In the same time, the other composite sample StMAPLE(266 nm)/Mg2.5AlPLD(1064 nm) with smaller roughness and different morphological aspect (Table 2, Figure 4) reveals a similar transition to hydrophobicity after long-time storage. The deposition of the stearic acid via MAPLE and of the Mg2.5Al-LDH occurred simultaneously; thus, the resulting composite films consist more likely in very disperse SA small aggregated embedded between the LDH layers.

The FT-IR spectra were recorded on films after deposition and after 36 months of storage. To scrutinize the behavior of the organic stearic acid component, the NIST standard FT-IR of the solid stearic acid is superimposed. The are some fingerprint peaks in the spectrum of the NIST solid stearic acid to survey in the spectrum of the as-deposited and stored films: the adsorption peaks at about 2915 cm^−1^ and 2850 cm^−1^, which are attributed to C-H asymmetric and symmetric stretching vibrations and the peak around 1702 cm^−1^ results from free carbon COO band [21]. Actually, C-H bands contain peaks corresponding to methyl (CH_3_-) and methylene (-CH_2_-). Both -CH_3_ and -CH_2_- are hydrophobic groups and have low surface energy decreasing the surface energy of the surface they cover or of the materials they are grafted to [35,36]. The free -COOH band of solid stearic acid is still present in the spectrum of the as-deposited StMAPLE(266 nm) film while it vanishes in the spectrum of the film after 36 months of storage, which shows that the stearic acid had reacted with the Si surface, although there are no changes in contact angle value [37,38]. It has been observed that an “ordered” film of alkanoic acids is characterized by alkyl chains in all-trans configurations arranged in such a way that they are tilted from the normal surface at the same angle. This results in νCH_2_-asymmetric ≤ 2916 cm^−1^ [37,38]. Table 3 includes a column with the position of this peak. The orientation of the alkane chains relative to surface normal may be surveyed through the relative intensity of ν_a_CH_2_ to the ν_a_CH_3_ bands, lower ν_a_CH_2_/ν_a_CH_3_ ratios being indicative of an average molecular orientation closer to the normal [39,40]. These values are included also in Table 3. The value of this ratio is quite high on as-deposited StMAPLE(266 nm) and decreases after long time storage marking an improving in the disorder in tilt in methyl-terminated alkane chains and yet, with no improving of the contact angle probably due to a lock of micro-nano structured surfaces. The observation of the evolution of steric interaction in the composite films is more complicated. For both as-deposited composite films, the peaks related to stearic acid are not defined and appears as a very large shoulder in the asymmetric and symmetric C-H stretching vibrations domain and an extremely broad peak in the asymmetric and symmetric COO- stretching vibrations. The 1702 cm^−1^ peak is not clearly observable, which indicates an interaction of stearic acid of extremely disoriented alkane chains embedded between oriented LDH crystallites. After 36 months of storage, in the FT-IR spectra of the composite films (Stearic acid-LDH), asymmetric and symmetric C-H stretching vibrations are visible and the free COO- peak at 1702 cm^−1^ is not perceived as indicative of an interaction of LDH matrix and stearic anions. The ν_a_CH_2_/ν_a_CH_3_ ratios are relatively high, which indicates a degree of freedom of swing and rearrangement of alkyl chains in spite the constraint of LDH matrix. The reorganization of stearic anions could explain the significant increase of CA in time, the surface becoming highly hydrophobic.

The examination of the peaks associated to LDH component is concentrated around the band at about 1460 cm^−1^, assignable to the symmetric and asymmetric stretching modes of CO_3_^2-^ and the very broad absorbance band in the range of 2800–3800 cm^−1^, generated by the stretching of H-bonded OH groups [41,42]. The CO_3_^2−^ band is primarily due to interlayer carbonates or introduced by CO_2_ from air. From Figure 6c, we observe an intense and well define band for both films Mg2.5AlPLD (1064 nm) and StMAPLE (266 nm)/Mg2.5AlPLD (1064 nm), which are consistent with the XRD results giving higher coherence lengths, D_003_, for these two samples. Upon 36 months of storage, the composite film exhibits almost no modifications standing for the time-stability of the LDH component. For the Mg2.5AlPLD (266 nm) and StMAPLE(266 nm)/Mg2.5AlPLD(266 nm) (Figure 6b), a slight increase of the intensity of this band for the composite film is observed, consistent with the XRD patterns exhibiting a better crystallinity of the composite film in comparison with its Mg2.5AlPLD (266 nm) reference. The stability of the cationic carbonate composition is also observed for this composite film. The broad absorbance band at 2800–3800 cm^−1^ is due to three main absorption bands: M-OH in the layer and interlayer, H_2_O-H_2_O bridging mode, hydrogen bonding of water in highly structured environment in interlayer space, and to CO_3_^2^-H_2_O stretching vibrations, water molecules hydrogen bonded to the carbonate ions present in the interlayer, appearing as a shoulder of this broad band. These three main absorption vibrations could be deconvoluted using a Gaussian in order to observe the modifications upon long storage times [42]. The results are included in a column in Table 3. The examination of the results included in Table 3 allow to emphasize the following aspects: the predominance of H_2_O-H_2_O bridges vibrations for the Mg, Al-LDH films for both 266 nm and 1064 nm deposition showing the affinity of LDH films for water. In the case of as-deposited composite films, the values of tail-band CO_3_^2^-H are overestimated due to the superposition to C-H stretching bands associated to the “disordered” stearic acid. However, the percentage of H_2_O-H_2_O bridges is less important. Upon long time storage, the CO_3_^2^-H disappears in the benefit of OH-M and H_2_O-H_2_O vibrations, due to the orientation of the stearic chains.

## 4. Conclusions

Composite organic-inorganic (stearic acid/Mg, Al-LDH) films were obtained by using a set-up combining the standard PLD and MAPLE. A MAPLE deposition system was adapted to provide the possibility of irradiating two concentric targets, with two different laser beams. The composite films exhibit long-term behavior different from the inorganic LDH films deposited by PLD and the stearic acid film deposited by MAPLE. A shift to hydrophobicity after 36 months of storage was observed for both composite films, although they display different morphologies and roughness. The result was explained, based on FT-IR measurements, on the reorientation in time of the alkyl chains of the highly dispersed stearic-acid aggregates embedded between LDH oriented layers. This particular structure is due to the simultaneous deposition of both the inorganic and the organic component. Such films exhibiting long-term stability and enhancement of their hydrophobicity are good candidates to be used as protective coatings. The combined PLD-MAPLE setup displays a large versatility in terms of ability of producing composite complex films or organic-inorganic heterostructures and sandwiches without opening the deposition chamber, and by this, reducing the contamination.

## Figures and Tables

**Figure 1 molecules-25-04097-f001:**
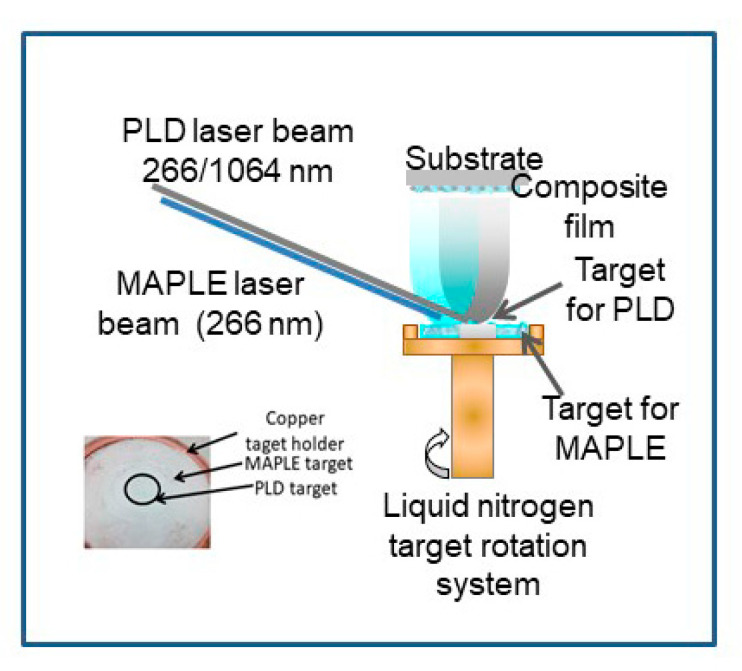
Experimental setup (combined matrix-assisted pulsed laser evaporation and pulsed laser deposition (MAPLE-PLD)) for thin films deposition.

**Figure 2 molecules-25-04097-f002:**
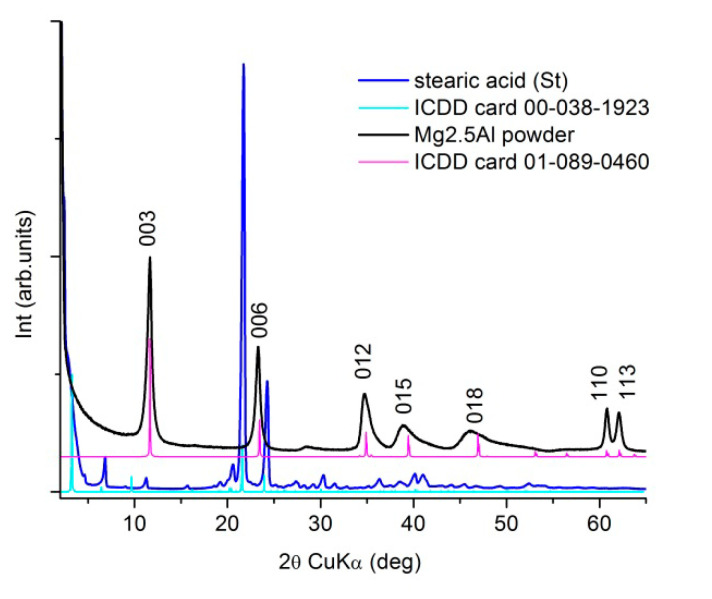
XRD patterns of the primary materials used for the targets’ preparation.

**Figure 3 molecules-25-04097-f003:**
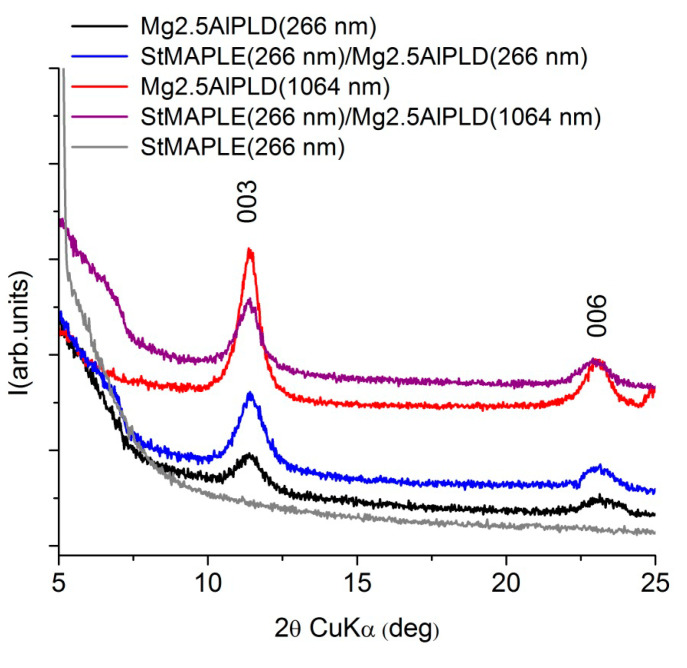
XRD patterns of the as-deposited films.

**Figure 4 molecules-25-04097-f004:**
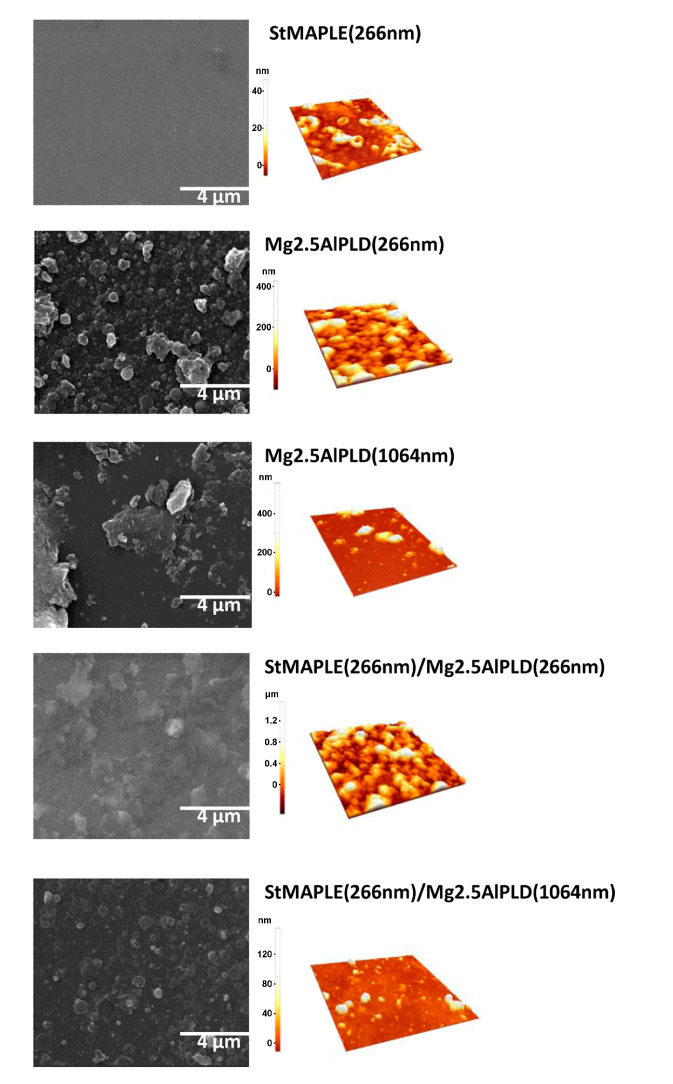
SEM images and AFM topography of 5 × 5 μm^2^ area of as-deposited films.

**Figure 5 molecules-25-04097-f005:**
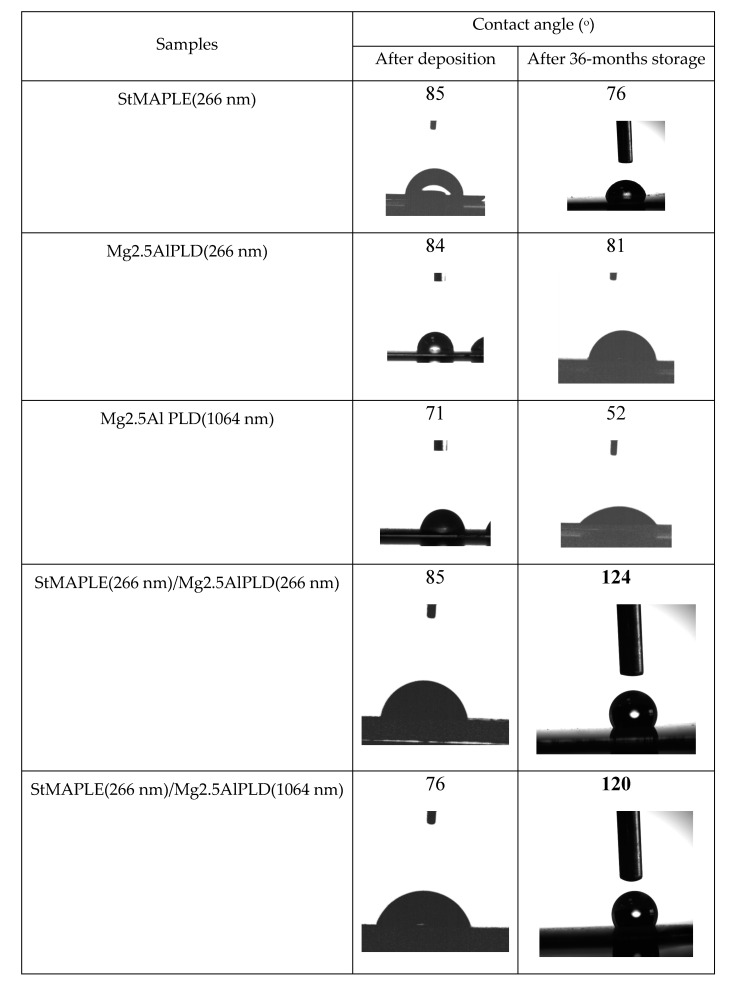
The contact angles values and the photographs of the water droplets on the surface of the as-deposited films.

**Figure 6 molecules-25-04097-f006:**
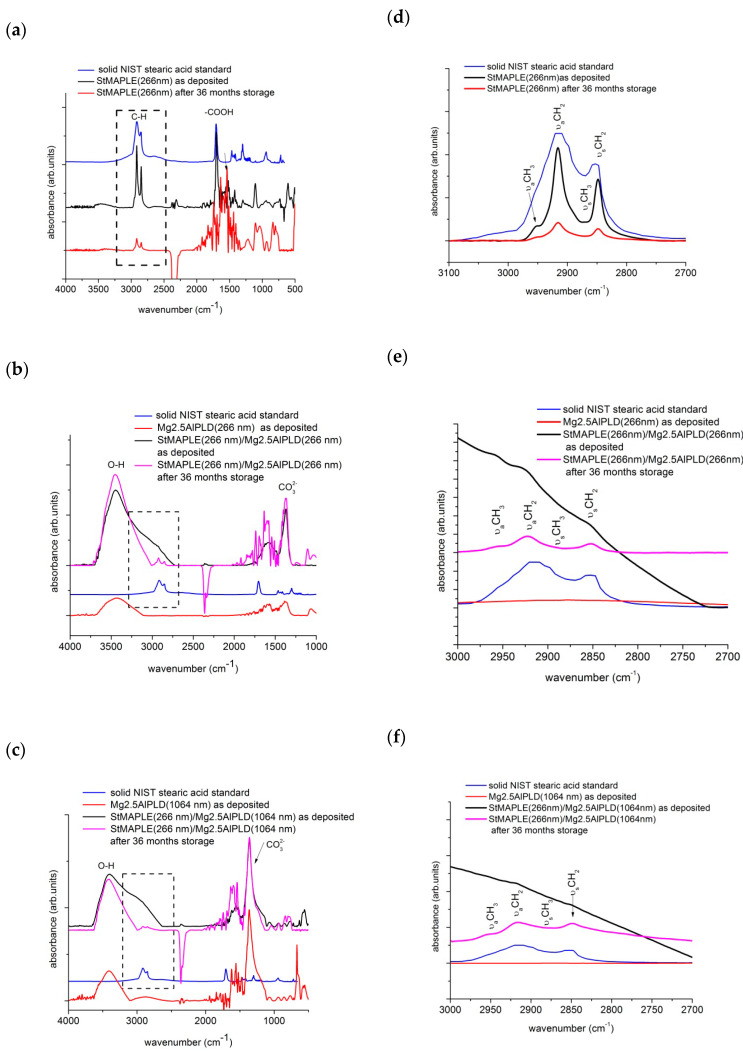
FT-IR spectra of as deposited films, of the stearic acid deposited via MAPLE (**a**) and of the composite films deposited via a combined MAPLE-PLD deposition (**b**) and (**c**). The FT-IR spectrum of the solid NIST (National Institute of Standards and Technology-US) is included. In the right column, the detailed spectra from 3000–2700 cm^−1^ wavenumbers are exposed, emphasizing the ν_a_CH_2_ and the ν_a_CH_3_ bands (**d**), (**e**) and (**f**), respectively.

**Table 1 molecules-25-04097-t001:** The codes used for the films and their corresponding deposition conditions.

Labels	Deposition Conditions
StMAPLE(266 nm)	Film of Pure Stearic Acid Deposited by MAPLE, at 266 nm Wavelength
Mg2.5AlPLD(266 nm)	Films of pristine Mg2.5Al deposited by standard PLD, at 266 nm wavelength
Mg2.5AlPLD(1064 nm)	Films of pristine Mg2.5Al deposited by standard PLD, at 1064 nm wavelength
StMAPLE (266 nm)/Mg2.5AlPLD(266 nm)	Stearic acid/layered double hydroxide composite thin films deposited by combined MAPLE-PLD: MAPLE at 266 nm and PLD at 266 nm wavelength
StMAPLE (266 nm)/Mg2.5AlPLD(1064 nm)	Stearic acid/layered double hydroxide composite thin films deposited by combined MAPLE-PLD: MAPLE at 266 nm and PLD at 1064 nm wavelength

**Table 2 molecules-25-04097-t002:** Structural data derived from XRD analysis and the roughness defined through the root mean square (RMS) deviations from AFM measurements.

Samples	Structural Data		RMS (nm)
StMAPLE(266 nm)	amorphous		8
	*c-*oriented LDH		
	*c* (Å)	D_003_ (nm)	
Mg2.5AlPLD(266 nm)	23.355	8.9	80
Mg2.5AlPLD(1064 nm)	23.303	10.0	23
StMAPLE (266 nm)/Mg2.5AlPLD(266 nm)	23.188	7.7	102
StMAPLE (266 nm)/Mg2.5AlPLD(1064 nm)	23.262	10.1	18

**Table 3 molecules-25-04097-t003:** FT-IR analyses of the as-deposited films.

Samples	Time	O-H Vibrations of LDH Component			C-H of Stearic Acid Vibrations	
		OH-M	H_2_O-H_2_O bridges	CO_3_^2-^-H	ν_a_CH_2_	ν_a_CH_2_/ν_a_CH_3_
Stearic acid NIST standard					2915	0.9
StMAPLE (266 nm)	as-deposited				2916	17.02
	36 months storage				2916	2.62
Mg2.5AlPLD(266 nm)	as-deposited	3561 cm^−1^ (0.04%)	3411 cm^−1^ (0.83%)	3230 cm ^−1^ (0.13%)		
StMAPLE(266 nm)/Mg2.5AlPLD(266 nm)	as-deposited	3488 cm^−1^ (0.31%)	3342 cm^−1^ (0.26%)	3055cm^−1^ (0.43%)		
	36 months storage	3474 cm^−1^ (0.54%)	3286 cm^−1^ (0.46 %)	-	2923	10.76
Mg2.5AlPLD(1064 nm)	as-deposited	3571 cm^−1^ (0.13%)	3426 cm^−1^ (0.70%)	3247 cm ^−1^ (0.17%)		
StMAPLE9266 nm)/Mg2.5AlPLD(1064 nm)	as-deposited	3461 cm^−1^ (0.20%)	3318 cm^−1^ (0.25%)	3023 cm ^−1^ (0.55%)		
	36 months storage	3445 cm^−1^ (0.45%)	3271 cm^−1^ (0.55%)	-	2917	10.25

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
