# Peer review of "Stearic Acid/Layered Double Hydroxides Composite Thin Films Deposited by Combined Laser Techniques"

_molecules, 2020, doi:10.3390/molecules25184097_

Round 1

Reviewer 1 Report

The typescript by R. Birjega et al. addresses an important issue of using new materials with a broad area of practical application. The subject is interesting , but I suggest to the authors to clear several issues, before publication of their work:

  1. The abstract part needs to be rewritten with a special focus on the aim of the work (explain why you obtain this type of thin films- short information).
  2. Could you correct: line 86 should be 400C and line 110, should be 2 J/cm2.
  3. The structure of thin film was described in introduction, but will be more clear when you put a new figure with the scheme of this structure.
  4. The resolution of figure 4 is very low, so please adjust (could you remove "4 um" from the thirds scheme of figure 4).
  5. The Conclusion part needs to be rewritten.

Reviewer 2 Report

Authors need to detail more about their research originality.

Is the process they used unique ? How so ? The use of PLD process to fabricate thin films is extensive, therefore authors need to clarify the originality of their work.

Did the authors analyze the cross sections of the fabricated composite films?

Figure 5 shows the contact angle property variation of the made films. What is the significance of the shown data ?  

Figure 6 is poorly constructed with inadequate cations. 

What is the conclusion of this work ? Composite LDH can be made ? This conclusion is too brief, it merely states the feasibility of the experiment tool the authors have used, it appears to be a brief research report conclusion. 

Round 2

Reviewer 2 Report

Revision of paper is deemed to be adequate